# Ribavirin as a First Treatment Approach for Hepatitis E Virus Infection in Transplant Recipient Patients

**DOI:** 10.3390/microorganisms8010051

**Published:** 2019-12-26

**Authors:** Antonio Rivero-Juarez, Nicolau Vallejo, Pedro Lopez-Lopez, Ana Isabel Díaz-Mareque, Mario Frias, Aldara Vallejo, Javier Caballero-Gómez, María Rodríguez-Velasco, Esther Molina, Antonio Aguilera

**Affiliations:** 1Infectious Diseases Unit, Clinical Virology and Zoonoses research group, Hospital Universitario Reina Sofía de Córdoba, Instituto Maimonides de Investigación Biomédica de Córdoba (IMIBIC), Universidad de Córdoba, 14006 Cordoba, Spain; lopezlopezpedro07@gmail.com (P.L.-L.); mariofriascasas@hotmail.com (M.F.); javiercaballero15@gmail.com (J.C.-G.); 2Digestive Unit, Complexo Hospitalario Universitario de Santiago, 15705 Santiago de Compostela, Spain; nicolau.vallejo.senra@sergas.es (N.V.); esther.molina.perez@sergas.es (E.M.); 3Nephrology Unit, Complexo Hospitalario Universitario de Santiago, 15705 Santiago de Compostela, Spain; ana.isabel.diaz.mareque@sergas.es; 4Microbiology Unit, Complexo Hospitalario Universitario de Santiago, University of Santiago de Compostela, 15705 Santiago de Compostela, Spain; laramva@hotmail.com (A.V.); maria.rodriguez.velasco@sergas.es (M.R.-V.); antonio.aguilera.guirao@sergas.es (A.A.); 5Animal Health Department, University of Cordoba-Agrifood Excellence International Campus (ceiA3), 15705 Cordoba, Spain

**Keywords:** HEV, treatment, zoonoses, ribavirin, transplant

## Abstract

The hepatitis E virus (HEV) is the major cause of acute hepatitis of viral origin worldwide. Despite its usual course as an asymptomatic self-limited hepatitis, there are highly susceptible populations, such as those with underlying immunosuppression, which could develop chronic hepatitis. In this situation, implementation of therapy is mandatory in the sense to facilitate viral clearance. Currently, there are no specific drugs approved for HEV infection, but ribavirin (RBV), the drug of choice, is used for off-label treatment. Here, we present two cases of chronic HEV infection in transplant patients, reviewing and discussing the therapeutic approach available in the literature. The use of RBV for the treatment of an HEV infection in organ transplant patients seems to be effective. The recommendation of 12 weeks of therapy is adequate in terms of efficacy. Nevertheless, there are important issues that urgently need to be assessed, such as optimal duration of therapy and drug dosage.

## 1. Introduction

Hepatitis E virus (HEV) is an RNA virus belonging to the *Orthohepevirus* genus that can infect a large number of hosts, including mammals (genera A, C and D) and birds (genus B) [1,2]. Viruses included in the Orthohepeviridae A family, genotypes 1 to 4, are a major cause of acute hepatitis worldwide [3]. The seroprevalence strongly varies between countries, become higher than 40% in Asia [3]. Moreover, genotypes 1 and 2 produce large outbreaks of acute hepatitis in low to middle income countries in Asia and Africa, and are associated with the consumption of contaminated water [4], whereas genotypes 3 and 4 have a global distribution producing sporadic cases, which are mainly documented in countries from the Americas and Europe, and are associated with the consumption of raw or undercooked food of animal origin [5,6]. Despite these two epidemiologically well-differentiated types of HEV that usually present as asymptomatic or self-limited acute hepatitis, clinical presentations and outcomes can vary depending on the population affected [7]. Genotypes 1 and 2 have been shown to have the worst prognosis, including a fatal outcome, in patients with underlying liver cirrhosis and pregnant women [8,9], while genotypes 3 and 4 can cause extrahepatic manifestations (highlighting those affecting central and peripheral nervous systems) and, under immunosuppression conditions, could involve chronic hepatitis [10,11].

Chronic HEV infection is defined as viral persistence in peripheral blood for more than three months [12,13]. The susceptible population comprises populations of patients with underlying immunosuppression, such as patients who are transplant recipients, patients who are HIV infected (with a total CD4^+^ cell count lower than 200 cells/mL) and patients treated with rituximab or with antitumoral necrosis factor drugs [12,13]. Once a chronic HEV infection is diagnosed, the therapeutic approach first consists of a reduction in the immunosuppression (if possible) and, in case of failure of self-resolution, implementation of therapy. Currently, there are no specific drugs approved for HEV infection, but ribavirin (RBV), the drug of choice, is used for off-label treatment. Studies investigating the safety and efficacy of the treatment of chronic HEV with RBV are limited to reports of medium to small cohorts and case reports, and clinical trials are lacking. Here, we present two cases of chronic HEV infection in transplant patients, reviewing and discussing the evidences for their first therapeutic approach.

## 2. Description of Cases

### 2.1. Case 1

A 65-year-old male who received a liver transplant in 2010 was admitted in February 2019 due to an elevated level of transaminases. The patient presented negative HCV antibodies, sHbAg and anti HBc, as well as HCV RNA and HBV DNA. In March, the patient was diagnosed with an HEV infection with detectable ant-HEV IgG and IgM antibodies and detectable HEV RNA in the serum. HEV antibodies were tested by indirective chemiluminescence immunoassay (CLIA) (Hepatitis E Virclia IgG Monotest and Hepatitis E Virclia IgM Monotest), following the manufacturer’s instructions. For HEV RNA determination, we used LightMix Modular Hepatitis E Virus (TIB MOLBIOL, Berlin, Germany). This assay has a detection limit set up at 10 genome equivalent copies or less per reaction (in vitro transcribed RNA). Total RNA extraction was performed from 200 µL of serum, using the automated MagNA Pure Compact RNA Isolation kit (Roche Diagnostics Corporation, Indianapolis, IN, USA), following the manufacturer’s instructions. The purified RNA was eluted in a total elution volume of 50 µL. The patient was receiving immunosuppressive therapy everolimus (0.5 mg every 12 h) in combination with mycophenolate mofetil (500 mg every 12 h). For phylogenetic analysis, a nested RT-PCR was performed, targeting the ORF2 region (structural proteins), using primers HEV_5920S (5’-CAAGGHTGGCGYTCKGTTGAGAC-3´) and HEV_6425A (5´-CAAGGHTGGCGYTCKGTTGAGAC-3´) in the first round and HEV_5930S (5´-GYTCKGTTGAGACCWCBGGBGT-3´) and HEV_6334A (5´-TTMACWGTRGCTCGCCATTGGC-3´) in the second round. The second amplification product of 467 bp was sequenced using the BigDye Terminator Cycle Sequencing Ready Reaction Kit on an ABI PRISM 3100 Genetic Analyzer (Applied Biosystems, Foster City, CA, USA). SnapGene software (Version 3.1; GSL Bio-tech, snapgene.com) was used for sequence analysis. The consensus sequence was obtained using SeqMan Software SeqMan NGen^®^ Version 12.0 (DNASTAR. Madison, WI, USA). Subtype assignment and phylogenetic analyses were performed using the HEVnet genotyping tool (https://www.rivm.nl/mpf/typingtool/hev/) and confirmed by BLAST. Genotyping of the viral strain was consistent with the genotype 3f. The patient reported, as risk factors for acquiring HEV, consumption of domestically prepared pork and wild boar. Twelve weeks after diagnosis of HEV infection, the patient remained with a detectable HEV viral load, been diagnosed with chronic infection. At this point, RBV was initiated at a dose of 200 mg every 8 h for 12 weeks, following clinical guidelines. During therapy, the patient experienced normalization of transaminases and did not report side effects associated with the medication. At the end of therapy, the HEV viral load was negative in the plasma and feces. The patient continues to have a negative viral load in the plasma and feces 24 weeks after completion of the therapy and has attained sustained virological response (SVR).

### 2.2. Case 2

A 35-year-old female had a kidney transplant in 2013. After transplantation, the patient experienced acute dysfunction of the graft and drug-related leucopenia was solved. From May 2018, the patient received prednisone (5 mg/day), tacrolimus (1.5 mg/day) and sirolimus (1 mg/day) as immunosuppressive therapy. The patient presented with an elevated level of transaminases in May 2018 and was suspected to have a drug-induced liver injury due to paracetamol, hypolipemiant therapy and contraceptive medication. She was negative for HCV antibodies, sHbAg and anti HBc, but positive for anti-HBs (48 mIU/mL). Viral load for HCV and HBV were also undetectable. Nevertheless, after drug interruption or switching, the elevated level of transaminases persisted. In March 2019, the patient was transferred to the hepatology unit and was diagnosed with an HEV infection, showing positive levels of anti-HEV IgG and IgM antibodies. HEV RNA was detected in the plasma and was determined to be genotype 3f. HEV RNA remains to be detectable 12 weeks later, confirming chronic HEV infection. As in the first case, RBV was initiated at a dose of 200 mg every 8 h for 12 weeks. During the therapy, the patient experienced normalization of transaminases and did not report side effects associated with the medication. At the end of therapy, the HEV viral load was negative in the plasma and feces, and the patient continues to have a negative viral load 24 weeks after the completion of therapy.

### 2.3. Ethics

This study was designed and performed according to the Helsinki Declaration. Patients provided permission to publish clinical data anonymously.

## 3. Discussion

After the diagnosis of a chronic HEV infection in organ transplant patients, clinical guidelines propose the first approach to reduce immunosuppression is to facilitate a self-limiting infection. A large series of 56 chronic HEV-infected transplant patients were evaluated, and the treatment strategy was to reduce the dose of the immunosuppressive medications [11]. Eighteen (32.1%) patients experienced viral clearance, with a median of 19.5 (10–106) months. Similarly, in a study in which 15 solid organ transplant patients were retrospectively identified with a chronic HEV infection, three experienced spontaneous viral clearance after the reduction in immunosuppressive therapy [14]. The resolving infection occurred between 3 and 30 months after initiating the reduction. In another cohort that included six chronically infected patients in whom the decreased dose was specifically aimed at facilitating HEV clearance, the infection was resolved in three patients (50%) [15]. In another study, in which five heart transplant recipients were diagnosed with chronic HEV infection, one patient experienced viral resolution after dose reduction of tacrolimus [16]. Thus, these relatively large cohorts show that a high proportion of patients could benefit from a dose reduction of immunosuppressive medications. Nevertheless, this result should be taken with caution. First, other studies or case reports showed a failure of viral clearance after immunosuppressive dose reduction [17,18,19,20,21]. Second, the elapsed time between the initiation of immunosuppression reduction and the time of self-resolution could be as high as or more than 1 year [11,14]. Finally, this reduction may be avoided in allogeneic hematopoietic stem cell transplant (alloHSCT) recipients, since a recent study found a strong association between immunosuppressive dose reduction and risk of mortality [22]. In this study, 2 out of 7 alloHSCT patients with a chronic HEV infection, in whom immunosuppressive drugs were reduced, had died due to the fulminant graft versus host disease. The identification of factors associated with failure/experience self-viral resolution can be helpful in terms of clinical decision making. In this sense, the use of tacrolimus rather than cyclosporin A has been identified as an independent predictive factor for the development of chronic HEV infection [11]; therefore, switching medications could be considered in these patients to facilitate HEV infection resolution. Nevertheless, cyclosporin A has been shown to have a higher risk of acute rejection than tacrolimus. For these reasons, reducing or changing immunosuppressive medications should be considered individually in the risk–benefit equation. For these reasons, the first therapeutic approach for inducing viral clearance in our cases was not the reduction of the immunosuppression therapy, but the uptake of RBV therapy. 

Despite the fact that the use of RBV for HEV infection should be considered as an experimental therapy due to its off-label use, a large number of cases reported in the literature use this compound, and RBV is recommended in clinical guidelines for HEV management in the general population, specifically in transplant-recipient individuals (Table 1) [12,13].

All these recommendations are based on a series of cases and case reports, summarized in Table 2. Because there are no clinical trials evaluating the proper course of RBV for the treatment of an HEV infection, the duration and dosage vary among studies (Table 2). Because of the disparity between studies, a meta-analysis cannot be conducted. However, considering the population reported as a whole, a total of 374 transplant patients have been treated with RBV and 276 of these patients have attained SVR, with a clearance rate of 73.7% (Table 2). When analyzing the treatment response by transplant organ, all populations included seemed to have a similar successful treatment outcome (kidney, liver, heart, lung and alloHSCT). Nevertheless, these data were cautiously reviewed because treatment lengths and drug dosages were variable. In this sense, clinical guidelines recommend a program therapy duration of 12 weeks using RBV at the initiation with a weight-adjusted dose or a dose adjusted on the basis of the estimated glomerular filtration rate [12,13,23]. Following these recommendations, a total of 176 transplant patients have been reported, and 129 of these patients have achieved SVR (73.2%) [11,24,25,26,27,28,29,30,31,32,33]. Therefore, this recommendation seems to be highly effective for the treatment of HEV infection in transplant recipients infected by genotype 3 HEV. Consequently, we decided to initiate this program in both cases following this recommendation. HEV was successfully cleared, showing a high safety and efficacy of this strategy in this setting. Despite that, a clinical trial evaluating the proper duration and dosage of RBV in transplant patients is mandatory.

However, despite the reported efficacy of RBV, there are several patients in who RBV is contraindicated, including pregnancy or those with a creatinine clearance lower than 50 mL/min [59]. Consequently, the development of new drugs is mandatory for the management of HEV infection. Therapeutic alternatives have been evaluated in vitro and in vivo. Sofosbuvir (SOF), a drug targeting the HCV NS5B polymerase, has shown to have an inhibitory effect on HEV replication [60]. Furthermore, the combination of SOF and RBV have demonstrated a cumulative effect for inhibiting HEV viral replication, suggesting a potential alternative approach. Nevertheless, its application in a clinical setting is inconclusive. While several studies have reported a successful treatment outcome of the combination of SOF + RBV in transplant recipients with chronic HEV infection, including one case of kidney and other of alloHSCT [61,62,63], others have not demonstrated its efficacy in this population [45,64,65]. Due to that, SOF seems to not add antiviral activity to RBV in vivo. On the other hand, the use of SOF monotherapy in those patients who are not eligible for RBV use has demonstrated a lack of efficacy [66]. Furthermore, the lack of efficacy also has been reported in the second-line treatment approach in transplant-recipient patients [67]. Consequently, the use of SOF is not recommended for the therapy of HEV infection. Other drugs have demonstrated antiviral activity against HEV in vitro, such Deptropine or Ciprofloxacin [68,69]. Nevertheless, currently there are no evidences in vivo and further investigation in this sense are needed.

## 4. Conclusions

In our experience, the use of RBV for the treatment of an HEV infection in organ transplant patients is effective, and the recommendation of 12 weeks of therapy being adequate in terms of efficacy. Nevertheless, there are important issues that urgently need to be assessed, such as the optimal duration of therapy and drug dosage of RBV, as well the development of new drugs with a high safety profile.

## Figures and Tables

**Table 1 microorganisms-08-00051-t001:** Treatment recommendations for chronic hepatitis E virus infection according to clinical guidelines.

Recommendation	EASL	GeHEP/SEIMC	BTS
**Reference**	[13]	[12]	[23]
**Drug**	RBV	RBV	RBV
**Treatment duration**	12 weeks	12 weeks	12 weeks
**Dosage**	Not specified	Weight-adjusted	Not specified
**Monitoring HEV RNA during therapy**	At week 12	At weeks 4 and 12	At weeks 1, 4, 8 and 12
**Increase therapy duration after completion of 12 weeks**	Up to 24 weeks if detectable viral load at week 12 in plasma/serum or feces	Up to 24 weeks if detectable viral load at week 12 in plasma/serum or feces	Continue until 2 stools > 1 month apart are both negative or continue for 24 weeks
**Retreatment**	RBV for 24 weeks. If failure, consider treatment with Peg-IFN alpha for 12 weeks *	RBV for 24 weeks. If failure, consider treatment with Peg-IFN alpha for 12 weeks *	RBV for 24 weeks. If failure, consider treatment with Peg-IFN alpha for 12 weeks *

European Association for the Study of the Liver (EASL); Grupo de Estudio de Hepatitis Virales (GeHEP) de la Sociedad Española de Enfermedades Infecciosas y Microbiología Clínica (SEIMC); British Transplantation Society (BTS); ribavirin (RBV); hepatitis E virus (HEV); ribonucleic acid (RNA); pegylated interferon (Peg-IFN). * Only should be considered in liver transplant patients.

**Table 2 microorganisms-08-00051-t002:** Efficacy of ribavirin monotherapy as first treatment for hepatitis E virus genotype 3 infection in adult transplant recipient patients.

Reference	*n*	Transplant	Country	Genotype	Duration, Median (Range) or Mean (SD)	Dosage	SVR (%) *	Note
**Studies on kidney transplant patients**								
Present study	1	Kidney	Spain	3	12 weeks	600 mg/day	1 (100)	
[24]	2	Kidney	Japan	3	12 weeks	600 mg/day	2 (100)	
[25]	16	Kidney	Germany	3	12 weeks	600 mg/day (200–800 mg)	15 (93.7)	Patients started therapy within the first 2 weeks after diagnosis of HEV infection
[26]	1	Kidney	France	3	12 weeks	1200 mg/day	1 (100)	Patient started therapy at the same week of diagnosis due to acute graft rejection
[34]	16	Kidney	Germany	3	15.2 (11.6)	85.7 mg (200 mg, thrice a week) and 1000 mg/day	11 (68.5)	
[35]	1	Kidney	Argentina	3	16 weeks	1000 mg/day	1 (100)	
[36]	1	Kidney	France	3	68 weeks	10 mg/kg	1 (100)	
[27]	2	Kidney	Spain	Unknown	12 weeks	600–800 mg/day	2 (100)	
[37]	1	Kidney	The Netherlands	3	16 weeks	400–600 mg/day	0	
[14]	4	Kidney	Germany	3	20 weeks	600–1200 mg/day	4 (100)	
[11]	6	Kidney	France	3	12 weeks	800 mg/day	4 (66.6)	
[38]	1	Kidney pediatric	Germany	3	12 weeks	10 mg/kg	0	Withdrew at week 4 due to severe adverse events
[18]	4	Kidney pediatric	Germany	3	12 weeks	9.7 mg/kg (range: 3.6 to 15.4 mg/kg)	3 (75)	Patient who did not achieve SVR experienced viral relapse after 2 months of therapy
[39]	1	Kidney-pancreas	France	3	12 weeks	12 mg/kg	1 (100)	
**Studies on liver transplant recipients**								
Present study	1	Liver	Spain	3	12 weeks	600 mg/day	1 (100)	
[40]	1	Liver	Uruguay	3	9 weeks	1200 mg/day	1 (100)	
[41]	4	Liver	Portugal	3	24 weeks	800–1200 mg/day	4 (100)	
[20]	1	Liver	Japan	3	20 weeks	800 mg/day	1 (100)	
[42]	1	Liver	Japan	3	20 weeks	200 mg/day and gradually increased up to 600 mg/day	1 (100)	
[28]	4	Liver	Germany	3	12 weeks	600 mg/day (200–800)	3 (75)	
[43]	1	Liver	Australia	3	12 weeks	200 mg/day	1 (100)	
[44]	1	Liver	Germany	3	16 weeks	600 mg/day	1 (100)	
[14]	4	Liver	Germany	3	20 weeks	600–1200 mg/day	3 (75)	One death before completing treatment
[21]	1	Liver pediatric	Germany	3	24 weeks	400 mg/day	1 (100)	
**Studies on Heart transplant recipients**								
[45]	1	Heart	France	3	12 weeks	800 mg/day	0	
[46]	1	Heart	France	3	10 weeks	200 mg/day and gradually increased up to 400 mg/day	0	Patient deceased at the end of therapy with detectable viral load
[47]	1	Heart	Sweden	3	36 weeks	800 mg/day for 18 weeks and 1200 mg/day for 18 weeks	1 (100)	
[16]	4	Heart	The Netherlands	3	12–36 weeks	200–800 mg/day	3 (75)	
[48]	4	Heart	Germany	3	20 weeks	800 mg/day	3 (75)	
[49]	1	Heart	France	3	12 weeks	17 mg/kg	1 (100)	
**Studies on Lung transplant recipients**								
[50]	4	Lung	Germany	3	18 weeks	400–800 mg/day	2 (50)	One patient died at week 4 of therapy due to acute graft rejection
[51]	2	Lung	The Netherlands	3	16 weeks	400 mg/day	2 (100)	One patient showed detectable HEV RNA in stools at evaluation of SVR
[14]	3	Lung	Germany	3	20 weeks	600–1200 mg/day	2 (66.6)	One death before treatment completion
**Studies on alloHSCT**								
[52]	13	alloHSCT	France	3	12 weeks (3–49 weeks)	400–1000 mg/day	11 (84.6)	One death before treatment completion
[22]	8	alloHSCT	France, Germany, the Netherlands and Scotland	3	12 weeks (1–32 weeks)	10 mg/kg (range: 5 to 22 mg/kg)	7 (87.5)	Patients 3, 1 and 4 received therapy with viral shedding <12, 12–24, and >24 weeks, respectively
**Studies not reporting SVR per type of transplant**								
[29]	48	Kidney (*n* = 29) Liver (*n* = 13) Heart (*n* = 3) Lung (*n* = 1) Kidney/Pancreas (*n* = 2)	France	3	12 weeks	600 mg/day (600–800 mg/day)	38 (79.1)	
[30]	8	Kidney (*n* = 5) Liver (*n* = 2) Bone marrow (*n* = 1)	Singapore	3	12 weeks	600 mg/day (400–800 mg/day)	2 (25)	All kidney transplant recipients fail to respond to therapy
[53]	63	Kidney (*n* = 45) Liver (*n* = 10) Heart (*n* = 5) Lung (*n* = 3)	France	3	12 weeks (12–72 weeks)	600 mg/day (200–1200 mg/day)	42 (66.6)	40 patients previously included in Kamar et al. NEJM 2014
[31]	35	Kidney (*n* = 22) Liver (*n* = 8) Heart (*n* = 3) Lung (*n* = 1) Kidney/Pancreas (*n* = 1)	France	3	12 weeks	600 mg/day (200–1200 mg/day)	22 (62.8)	22 patients previously included in Kamar et al. NEJM 2014
[32]	4	Liver (*n* = 1) Kidney (*n* = 2) Liver/Kidney (*n* = 1)	Spain	3	12 weeks	600–800 mg/day	3 (75)	
[33]	24	Kidney (*n* = 16) Liver (*n* = 5) Heart (n = 2) Lung (*n* = 1)	France	3	12 weeks	600 mg/day (200–1200 mg/day)	15 (62.5)	
[54]	15	Not specified	Germany	3	Not specified	Not specified	13 (86.6)	
[55]	59	Kidney (*n* = 37) Liver (*n* = 10) Heart (*n* = 5) Lung (*n* = 2) Kidney/Pancreas (*n* = 5)	France	3	12 weeks (4–72 weeks)	600 mg/day (29–1200)	46 (77.9)	
[56]	41	Kidney (*n* = 26) Liver (*n* = 9) Heart (*n* = 3) Kidney/pancreas (*n* = 1) Lung (*n* = 1) Liver/kidney (*n* = 1)	France	3	12 weeks	9.7 mg/kg/day (2.7–16.3)	25 (61%)	
**Studies on transplant patients infected by HEV genotype different than 3**								
[19]	1	Kidney	China	4	24 weeks	RBV (not specified)	0	
[57]	3	Kidney	China	4	12 weeks	RBV (800 mg/day)	2 (66.6)	
[17]	1	Liver	Switzerland	3ra	16 weeks RBV + 24 weeks RBV/SOF + 16 weeks RBV	RBV (1129–3700 ng/mL) and SOF (400 mg/day)	0	
[58]	1	Not specified	Switzerland	3ra	12 weeks	RBV (not specified)	1 (100)	
[30]	1	Liver	Singapore (patient from United Arab Emirates)	7	12 weeks	600 mg/day (400–800 mg/day)	1 (100)	

Number of patients (*n*); standard deviation (SD); sustained virological response (SVR); milligram (mg); hepatitis E virus (HEV); allogenic hematopoietic stem cell transplantation (alloHSCT). * SVR was defined as undetectable HEV RNA in serum and stools 12 and/or 24 weeks after completing the therapy.

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
