# Peer review of "Ribavirin as a First Treatment Approach for Hepatitis E Virus Infection in Transplant Recipient Patients"

_microorganisms, 2019, doi:10.3390/microorganisms8010051_

Round 1

Reviewer 1 Report

In the submitted manuscript entitled "Ribavirin as a first treatment approach for hepatitis E virus infection in transplant recipient patients: case reports and systematic review", Rivero-Juarez et al. discuss the use of ribavirin for treatment of two cases infected with HEV after organ, liver/kidney, transplantation. They also summarize most of the previous studies on ribavirin-based therapy in the context of organ transplantation. The study is interesting, since little is known about the efficiency of ribavirin for treatment of HEV in patients with organ transplantation, and the manuscript is well written, however, in my opinion, some points still need to be addressed. 

- In the current form of the manuscript, the authors described the two cases and the process of internet search in the result section whereas the materials and method section is missing. In my opinion, the structure of the manuscript should be changed to: Introduction, Materials and methods, Results and discussion. All details regarding the methodology should be mentioned in the materials and methods part.

- Also there is no information regarding the assays used for HEV detection and sequencing as well as genotyping. Please mention these in details and put the limit of quantification/detection of virus detection kit. Also how the virus was genotyped as 3f should be clarified.

- If there is any information regarding the presence/absence of any other hepatic viruses such as HCV or HBV in the two patients please mention.

- The sections Results and Discussion could be combined together, and in this section the authors still need discuss ribavirin-based therapy in details in relation to their two cases and to cite some extra work of ribavirin-based treatment in the context of solid organ transplantation. Examples of the related work:

Lhomme et al., 2019; van Wezel et al., 2019; Wahid B 2019; Donnelly MC et al., 2017; Nishiyama T et al., 2019; Lhomme et al., 2015; ….

-I missed any discussion on the current two treated cases (case 1 and case 2) in the discussion part?

-Also it will be beneficial to shed the light on any promising natural/synthetic compounds, if any, in addition to combination therapy using HCV inhibitors like sofosbuvir.

- In the introduction, please add a small paragraph on the biology of the virus including the worldwide prevalence, diagnosis, SVR of chronic HEV treatment, the symptoms and disease progression, etc.

Minor comments:

- Page 2 line 4, please make (+) in the CD4+ as superscript.

- Page 2 line 45 of case 2, please change 6 months to 24 weeks to match with case 1 line 30.

- Page 2 line 40, remove hepatitis E virus and put the abbreviated form (HEV).

- Page 2 line 29 and 44, mention the detection limit of the assay (check the above comment).

- Page 3 line 1, in the sentence “After the diagnosis of a chronic HEV infection”, do the authors mean in organ transplant setting or in chronic HEV generally. Please clarify?

- Page 3 line 32, the sentence “Currently, there are no clinical trials evaluating the 33 efficacy, optimal duration or dosage of RBV for the treatment of HEV” is repeated in line35. Please delete the first to avoid redundancy. Also mention the benefits of using ribavirin for the current/previous HCV-combination therapies.

 - Page 4 table 1, put BTS in the description part, I think it is missed.

- The authors should discuss the studies mentioned in the tables, or at least the most relevant ones, in the discussion part in relation to their two cases. Also discuss the possible reasons for the differences in the SVR among studies presented in the tables.

Author Response

In the submitted manuscript entitled "Ribavirin as a first treatment approach for hepatitis E virus infection in transplant recipient patients: case reports and systematic review", Rivero-Juarez et al. discuss the use of ribavirin for treatment of two cases infected with HEV after organ, liver/kidney, transplantation. They also summarize most of the previous studies on ribavirin-based therapy in the context of organ transplantation. The study is interesting, since little is known about the efficiency of ribavirin for treatment of HEV in patients with organ transplantation, and the manuscript is well written, however, in my opinion, some points still need to be addressed.

We would like to thank reviewer consideration with our work.

In the current form of the manuscript, the authors described the two cases and the process of internet search in the result section whereas the materials and method section is missing. In my opinion, the structure of the manuscript should be changed to: Introduction, Materials and methods, Results and discussion. All details regarding the methodology should be mentioned in the materials and methods part.

We have now changed the structure of the manuscript.

Also there is no information regarding the assays used for HEV detection and sequencing as well as genotyping. Please mention these in details and put the limit of quantification/detection of virus detection kit. Also how the virus was genotyped as 3f should be clarified.

We have included in section methods all the information regarding to HEV IgG and IgM determination as well PCR protocol for detection and sequencing.

If there is any information regarding the presence/absence of any other hepatic viruses such as HCV or HBV in the two patients please mention.

We have included in every cases information relative to HCV and HBV.

The sections Results and Discussion could be combined together, and in this section the authors still need discuss ribavirin-based therapy in details in relation to their two cases and to cite some extra work of ribavirin-based treatment in the context of solid organ transplantation. Examples of the related work: Lhomme et al., 2019; van Wezel et al., 2019; Wahid B 2019; Donnelly MC et al., 2017; Nishiyama T et al., 2019; Lhomme et al., 2015.

Due to the new structure of the manuscript, and following suggestion of reviewer 3, the discussion section now is a combination of discussion and a review. In this sense, we have included references suggested by reviewer among others.

I missed any discussion on the current two treated cases (case 1 and case 2) in the discussion part?

We have now included discussion of the cases.

Also it will be beneficial to shed the light on any promising natural/synthetic compounds, if any, in addition to combination therapy using HCV inhibitors like sofosbuvir.

We agree with reviewer that this information will be really useful for a potential reader. A new paragraph focusing in this point has been included.

In the introduction, please add a small paragraph on the biology of the virus including the worldwide prevalence, diagnosis, SVR of chronic HEV treatment, the symptoms and disease progression, etc.

We have now included in the introduction information regarding to seroprevalence, virology and symptoms of HEV.

Minor comments:

9.1. Page 2 line 4, please make (+) in the CD4+ as superscript.

Corrected

9.2. Page 2 line 45 of case 2, please change 6 months to 24 weeks to match with case 1 line 30.

Changed

9.3. Page 2 line 40, remove hepatitis E virus and put the abbreviated form (HEV).

Abbreviated

9.4. Page 2 line 29 and 44, mention the detection limit of the assay (check the above comment).

Included now in methods

9.5. Page 3 line 1, in the sentence “After the diagnosis of a chronic HEV infection”, do the authors mean in organ transplant setting or in chronic HEV generally. Please clarify?

It means to transplant population. We have now included it.

9.6 Page 3 line 32, the sentence “Currently, there are no clinical trials evaluating the 33 efficacy, optimal duration or dosage of RBV for the treatment of HEV” is repeated in line35. Please delete the first to avoid redundancy. Also mention the benefits of using ribavirin for the current/previous HCV-combination therapies.

We have deleted the sentence in sense to avoid redundancy.

9.7 Page 4 table 1, put BTS in the description part, I think it is missed.

BTS has been included in the description part

9.8 The authors should discuss the studies mentioned in the tables, or at least the most relevant ones, in the discussion part in relation to their two cases. Also discuss the possible reasons for the differences in the SVR among studies presented in the tables.

We have now expanded information about studies referenced. Nevertheless, due to the word count limit the number of studies that can be comment is limited.  

Reviewer 2 Report

In the present study, the reported cases with ribavirin treatment for HEV infection in transplant recipients are summarized.

Comments:

As for authors’ cases, it is not clear whether authors confirm HEV infection was persistent for more than 12 weeks. Did authors try to reduce the immunosuppressive therapy? How did authors determine the dose (weight-adjusted?) and duration of ribavirin for their cases?

In the case 2 patient, did acute dysfunction of graft and the elevated level of transaminases occur simultaneously in May 2018? Can authors confirm HEV infection at this time?

Can ribavirin be used for patients with severe liver dysfunction or severe renal dysfunction?

How common is HEV infection and is HEV RNA test popular in authors’ institution or country?

Author Response

As for authors’ cases, it is not clear whether authors confirm HEV infection was persistent for more than 12 weeks. Did authors try to reduce the immunosuppressive therapy? How did authors determine the dose (weight-adjusted?) and duration of ribavirin for their cases?

Both cases received RBV as first therapeutic approach without previous immunosuppressive therapy reduction. We have now included all this information in discussion and justify the reasons for this clinical decision.

In the case 2 patient, did acute dysfunction of graft and the elevated level of transaminases occur simultaneously in May 2018? Can authors confirm HEV infection at this time?

Yes, the dysfunction of the graft occurs simultaneously in May 2018. Unfortunatelly, HEV was not determined at this point and we do not have sample available at this point to evaluate it.

Can ribavirin be used for patients with severe liver dysfunction or severe renal dysfunction?

RBV cannot be administered in severe renal dysfunction (creatinine clearance lower than 50 mL/min). We have included in discussion a paragraph about contraindication of RBV and its limitations for its uptake in this setting.

How common is HEV infection and is HEV RNA test popular in authors’ institution or country?

There are not specific studies conducted in Galicia (North West Spain) about prevalence of HEV in general population or special populations (HIV, transplant recipient, cirrhotic…). In Spain the seroprevalence is estimated in 10%-15%, but this data strongly varies between regions and age.

Reviewer 3 Report

Drs Rivero-Juarez and colleagues have presented a per se interesting article (case report and review of the literature) in which they aimed to explore the use and efficacy of RBV for treatment of chronic hepatitis E (CHE). The colleagues have included in their case report two CHE cases under RBV treatment. As a result they could show the successful therapy strategy using RBV while the patients showed SVR after treatment. Subsequently, the authors tried to review the literature in a separate chapter.

Overall, the manuscript needs major revision and the following comments should be addressed.

Major compulsory comments

The English style and grammar and also the terminology have to be checked carefully best by a native (academic) English speaker. The article is not well structured. It is not clear to me, why the authors added the two cases of CHE patients under RBV treatment here. They briefly and only superficially describe these patients showing no clinical, virological, and demographic data, e.g. a table summarizing the patient and virological data are missing. Furthermore, there is no discussion in terms of the experience the authors have made from these two patients and a conclusion is missing. Therefore, the description of the two cases in the context of the article is confusing. An ethical approval for the cases is missing.

The colleagues stated in the abstract that they would review the research available in the literature. However, they didn´t introduce the review chapter and, more importantly, didn´t structure the review part by itself. For example the authors didn´t introduce a meta-analyses needed for such a review. They also didn´t subdivide the review section, e.g., highlighted new or important findings, and didn´t comment on the findings of their review. As a reader I am confused of the review section.

Author Response

The English style and grammar and also the terminology have to be checked carefully best by a native (academic) English speaker.

Both grammar and terminology have been revised by American Journal Expert.

The article is not well structured. It is not clear to me, why the authors added the two cases of CHE patients under RBV treatment here. They briefly and only superficially describe these patients showing no clinical, virological, and demographic data, e.g. a table summarizing the patient and virological data are missing. Furthermore, there is no discussion in terms of the experience the authors have made from these two patients and a conclusion is missing. Therefore, the description of the two cases in the context of the article is confusing. An ethical approval for the cases is missing. The colleagues stated in the abstract that they would review the research available in the literature. However, they didn´t introduce the review chapter and, more importantly, didn´t structure the review part by itself. For example, the authors didn´t introduce a meta-analyses needed for such a review. They also didn´t subdivide the review section, e.g., highlighted new or important findings, and didn´t comment on the findings of their review. As a reader I am confused of the review section.

We have restructured the manuscript following reviewer recommendation. Now the paper included a Methods section and information regarding cases has been expanded. Also we have included information regarding experience within our cases in discussion. Consequently, the manuscript is now more similar to an original investigation than to a review. We have also included a paragraph about new therapeutics perspectives.

Round 2

Reviewer 1 Report

In the resubmitted manuscript "Ribavirin as a first treatment approach for hepatitis E virus infection in transplant recipient patients: case reports and systematic review", the authors discuss ribavirin-based HEV therapy in the context of organ transplantation. Although the topic is of particular interest, the way the authors present their data is confusing and not convincing. The combination of case study and the literature review significantly weaken both compartments. In their future submission steps, the authors are advised to write it as a case report. Alternatively, a detailed review article for the topic will be also attractive for readers. 

Author Response

We have now submitted the manuscript as case report, and, following reviewer suggestion, prepare a new review manuscript to be submitted in the future.

Reviewer 3 Report

I have read the revised version of the manuscript and find that the authors have improved their review/case report. I have no further questions

Author Response

We would like to thanks reviewer 3 suggestions and comments.

Round 3

Reviewer 1 Report

In the submitted manuscript "Ribavirin as a first treatment approach for hepatitis E virus infection in transplant recipient patients: case reports and systematic review", the authors present two cases for ribavirin treatment after liver/kidney transplantation. They showed that ribavirin can be an effective option for treatment of these category of patients. Since the authors choose to present their data as a case report, they should follow the regular structure of this type of research. Normally, the structure of the case report is as following: abstract, Introduction,Case presentation, and Discussion. Based on this, the authors need to adjust their manuscript accordingly.

Specific comments:

Remove the material and methods section. Replace the title "results" by "Description of cases" or directly by "case 1" and "Case 2". In the cases section, clarify the time points for HEV and liver enzymes investigations. If you tested HEV and liver enzymes at different time points, please make a graph and upload. The kinetics of virus decline is important. In case 2, you replace "6 months" by "24", but months was not replaced by weeks. Please check and correct to 24 weeks" to match with case 1.

Author Response

We would like to thanks reviewer for their comments and suggestions. Following their recommendations, we have changed now the structure of the manuscript removing Material and Methods section. We have now integrated description of HEV antibodies, HEV-RNA determination and typing in Case 1. Also we have changed the Results section head to Description of cases. Finally, we have corrected "24 months", now "24 weeks". 

Unfortunately, we do not have additional HEV-RNA and liver enzymes determination (further than presented in the current version of the manuscript). Consequently, we cannot make a graph in this sense.